# Olympic combat sports and mental health in children and adolescents with disability: A protocol paper for systematic review

Youngjun Lee[1], Laura Capranica[2], Caterina Pesce[2], Flavia Guidotti[2,3], Valentin Benzing[4], Janet Hauck[1]*, Simone Ciaccioni[2]*

**1** Department of Kinesiology, Michigan State University, East Lansing, MI, United States of America,
**2** Department of Movement, Human and Health Sciences, University of Rome "Foro Italico", Rome, Italy,
**3** Department of Human Sciences and Promotion of the Quality of Life, San Raffaele Roma Open University, Rome, Italy, **4** Institute of Sport Science, University of Bern, Bern, Switzerland

* hauckja1@msu.edu (JH); simoneciaccioni@yahoo.it (SC)

## Abstract

### Introduction

Mental health is important for children and adolescents, particularly those with disabilities. While the mental health advantages of sports participation are well-documented, the specific type of sport may have heightened relevance for children and adolescents with disabilities. The objective of this systematic review protocol is to outline the rationale and methodology for investigating how participation in Olympic combat sports influences the mental health outcomes of this unique population, which is more susceptible to developing mental health issues than their neurotypical counterparts.

### Methods and analysis

A comprehensive search will be conducted across academic databases, including the Cochrane Library, ERIC, PsycINFO, PubMed, Scopus, SPORTDiscus, and Web of Science. The focus will be on identifying randomized and non-randomized controlled trials (RCTs and non-RCTs, respectively), and observational studies with control groups that explore the impact of Olympic combat sports on the mental health of children and adolescents with disabilities. To assess the risk of bias, the Rob 2.0 tool will be employed for RCTs, and the ROBINS-I tool for CTs. For longitudinal and cross-sectional studies, the National Institute of Health's Study Quality Assessment Tool for Observational Cohort and Cross-sectional Studies will be used. The review process will be conducted using Covidence, possibly utilizing JASP software for meta-analysis if the retrieved studies exhibit sufficient homogeneity. Data that cannot be included in the meta-analysis will be synthesized using the Synthesis without Meta-Analysis (SWiM) tool. Furthermore, the Consolidated Framework for Implementation Research (CFIR) will provide a framework consisting of five broad domains: intervention characteristics, outer setting, inner setting, characteristics of individuals, and the process of implementation.

**Data Availability Statement:** Deidentified research data will be made publicly available when the study is completed and published.

**Funding:** The author(s) received no specific funding for this work.

**Competing interests:** The authors have declared that no competing interests exist.

**Abbreviations:** BDI, Beck Depression Inventory; C-SSRS, Columbia-Suicide Severity Rating Scale; CFIR, Consolidated Framework for Implementation Research; CI, Confidence Intervals; CPSS, Childhood Perceived Stress Scale; CTP, California Test Personality; GAD-7, Generalized Anxiety Disorder 7 (GAD-7; GARS-2, Gilliam Autism Rating Scale-Second Edition; GHQ, General Health Questionnaire; K10, Kessler Psychological Distress Scale (K10; MA, Meta-analyses; OCS, Olympic combat sports; OHQ, Oxford Happiness Questionnaire; PHQ-9, Patient Health Questionnaire-9; PRISMA-P, Preferred Reporting Items for Systematic Review and Meta-analysis Protocols; RCTs, Randomized Controlled Trials; RoBMA, Robust Bayesian Meta-analysis; ROBINS-I, Risk Of Bias In Non-randomized Studies–of Interventions; SF-36, = Short Form 36 Health Survey; SMD, Standardized Mean Difference; SLRs, = Systematic Literature Reviews; STAI, State-Trait Anxiety Inventory; SWiM, Synthesis without Meta-Analysis; SWLS, = Satisfaction with Life Scale; WHOQOL, World Health Organization Quality of Life (WHOQOL.

## 1 Introduction

In the world of sports, the Olympic Games serve as a symbol of unity, excellence, and human achievement [1]. Within this global spectacle, we witness athletes from diverse backgrounds competing at the highest level. However, it is not just elite athletes on the world stage who benefit from sports. Olympic combat sports (OCS), including disciplines like boxing, fencing, judo, karate, taekwondo, and wrestling, have long been celebrated for their unique combination of physical prowess, technique, and mental acuity. These sports not only captivate audiences with their electrifying bouts and displays of athleticism but also offer a plethora of health benefits to practitioners [2, 3]. Since OCS demands rigorous cardiovascular conditioning due to the intermittent bursts of high-intensity activity intertwined with moments of strategic pacing, athletes in these disciplines often exhibit elevated levels of aerobic fitness, which contribute to improved heart health, reduced risk of cardiovascular diseases, and better overall endurance [4, 5]. Engaging in OCS requires the development of functional strength and muscular power, with training routines encompassing a blend of resistance exercises, bodyweight drills, and sport-specific movements [6]. Therefore, these activities promote muscular hypertrophy, endurance, and improved body composition, thereby fostering greater overall strength and physical resilience [7]. Moreover, the intricate techniques and dynamic movements inherent to OCS necessitate exceptional agility, balance, and coordination. Practitioners must execute precise actions, often in the heat of combat, leading to improvements in motor skills and proprioception [8].

These enhancements extend beyond the sporting arena and can have positive implications for everyday activities. In fact, under high-pressure situations practitioners learn to manage stress, maintain focus under duress, and develop resilience in the face of adversity, which are mental skills that can be applied to various aspects of life, promoting mental health [9, 10]. When not overstressed, OCS weight categories necessitate athletes to manage their body composition meticulously, encouraging healthy eating habits and weight management strategies, which can translate to better overall health and reduced risk of obesity-related issues [11, 12]. Furthermore, the impact forces experienced during training and competition in combat sports can contribute to increased bone density, reducing the risk of osteoporosis and fractures, particularly in aging populations [13]. Finally, engagement in OCS often involves being part of a close-knit community of practitioners and coaches. This social aspect fosters camaraderie, support networks, and a sense of belonging, which are essential for mental health and stress reduction [14, 15].

Underscoring the enduring appeal of OCS for individuals seeking to optimize their well-being through disciplined and engaging physical activity, OCS holistic advantages have also demonstrated significant potential to promote inclusivity for individuals with disabilities [16, 17]. What remains less explored, though, is the potential impact of these sports on the mental health, of children and adolescents living with disabilities. Mental health is a state of well-being and effective functioning in which an individual realizes his or her abilities [18]. Thus, improving and maintaining its function are especially important for individuals with developmental disabilities, which refer to conditions affecting the body or mind that can hinder certain activities and interactions with the surrounding environment [18]. This paper outlines a protocol for a systematic literature review aimed at exploring potential links between OCS and the mental health of this specific and often underserved group.

The importance of physical activity (i.e., any bodily movement produced by skeletal muscles that requires energy expenditure at any intensity; [19]), particularly in the form of sports, for individuals with disabilities cannot be overstated [20]. Participation in sports not only enhances physical health but also empowers individuals by fostering self-confidence, social

skills, and quality of life across the lifespan [21–23]. OCS, known for its emphasis on technique, strategy, and discipline, offers a particularly promising avenue for holistic development in children and adolescents with disabilities [24, 25]. By engaging in these sports, young individuals with disabilities can acquire valuable life skills, learn to overcome challenges and experience a sense of belonging as part of a team or community [26].

Despite the potential benefits, the relationship between OCS and the mental health of children and adolescents with disabilities remains largely unexplored in the academic literature. While existing studies have hinted at the positive effects of sports on the mental health of individuals with disabilities [27], a comprehensive systematic review is needed to further our understanding of the OCS-mental health relation in this special population. The aim of the present systematic review is twofold: (1) identifying which consistencies in results may allow to draw first conclusions with applied implications for children and adolescents with disabilities; and (2) which gaps call for future research. Specifically, this work aims to address potential gap by i) delving into the existing literature on the link between OCS participation and mental health among children and adolescents with disabilities with robust data extraction and risk of bias assessment processes; ii) establishing specific research questions and hypotheses; iii) specifying clear inclusion and exclusion criteria within selected studies, including study types, participants, interventions, comparators, and mental health outcomes. Outlining the framework for the research procedures, ensuring transparency, reproducibility, and scientific integrity, this protocol paper provides a clear roadmap, facilitating collaboration, and advances knowledge by guiding the planned systematic review [28].

## 2 Materials and methods

The present protocol has been registered in the International Prospective Register of Systematic Reviews (PROSPERO) with a registration number of CRD42023452489 on August 26, 2023. The formulation of this protocol manuscript adhered to the guidelines provided by the Preferred Reporting Items for Systematic Review and Meta-analysis Protocols (PRISMA-P) (S1 File) [29]. For experimental studies, the PICOS [30] framework will be employed to guide our research. In the context of observational studies, we will utilize the PECO [31] framework. To assess the impact of OCS intervention programs, subgroup moderator analyses will be carried out, taking into consideration various factors such as study duration/effect type (acute vs. chronic), age group (children vs. adolescents), type of intervention (if given), disability category, and exercise intensity. These subgroups will be loosely constructed based on the principles of the Consolidated Framework for Implementation Research (CFIR), encompassing aspects like program characteristics (including type, duration, and mode), location and setting, method of delivery, attributes of program providers, social context, delivery approach, policy considerations, individual characteristics of each special population served by the program, fidelity of implementation, program adoption, and adherence.

### 2.1 Population

We will consider studies encompassing children and adolescents aged 5–18 years who have disabilities in various developmental domains, emotions, intellect, physical abilities, and sensory functions for inclusion. The overall construct of disability signifies the adverse aspects of the interplay between an individual and their contextual factors. Studies that encompass children and/or adolescents with documented ongoing medical conditions that are known to impact their participation in physical activity and involve participants receiving treatment at all levels of care, will be excluded. This exclusion pertains to studies involving participants with conditions such as cancer or anterior cruciate ligament injuries, as well as studies in

which the intervention occurs in a clinical setting. Studies providing data for age groups beyond the specified age range will also be excluded unless they offer data specific to a sub-group falling within the eligible mean age.

## 2.2 Study design

We will incorporate studies that have undergone peer review, have been published in reputable journals and performed with a wide range of research designs, including observational studies with control groups and experimental intervention studies. Specifically, as regards experimental interventional studies, we will include RCTs and non-RCTs involving OCS programs specifically designed for children and adolescents with developmental, emotional, intellectual, physical, and sensory disabilities, as well as observational studies with a longitudinal non-interventional, or cross-sectional design that address the association of OCS practice with mental health in the target population. We will also explore interventions for children and adolescents, covering both transient and permanent disability conditions. These interventions may involve OCS-based studies, which can encompass the assessment of transient effects from a single acute bout of activity or longer-lasting effects resulting from prolonged, chronic OCS training. However, we will exclude qualitative studies, reviews, meta-analyses, reports, protocols, letters, editorials, and gray literature from our analysis.

The eligible study types for inclusion will encompass observational studies with control groups, RCTs, CTs, and studies that have undergone peer review and have been published in reputable journals.

## 2.3 Outcomes

The primary focus of this protocol centers on assessing mental health outcomes. Identifying key domains within mental health, such as mental health literacy, attitudes toward mental disorders, and social skills, can contribute to a more robust body of evidence, shedding light on pathways to enhance mental health [32].

Our principal outcome measures encompass various facets of mental health, categorized into several broader domains. These domains include psychological distress, severity of depression, severity of anxiety, quality of life, well-being and life satisfaction, stress levels, and suicidal ideation and behavior. To specify these domains further and address the challenge of handling data from various scales, we outline the following approach:

- Scale Reliability and Validity Assessment: Before analysis, we will review the latest literature to confirm the reliability and validity of each scale within our target population. This step is crucial for ensuring that our findings are grounded in reliable and valid measures.

- Normalization and Standardization of Scores: Given the different versions of scales and their varying total scores, we will apply normalization or standardization techniques where appropriate. This will allow us to compare scores across different scales effectively.

- Handling Non-Linear Score Distributions: For scales known to have non-linear score distributions, we will employ appropriate statistical methods, such as transformation techniques or non-parametric analyses. This will allow us to render linearly and non-linearly distributed data comparable, and avoid violation of analysis assumptions, thus ensuring an accurate interpretation of the results.

- Integrated Analysis Approach: Where possible, we will use meta-analytic techniques to integrate findings across different scales, considering their psychometric properties. This

integrated approach will help us synthesize evidence more effectively, providing a comprehensive view of the outcomes.

Outcome Measures and Scales:

### 2.3.1 Psychological distress.

- General Health Questionnaire (GHQ)
- Kessler Psychological Distress Scale (K10)

### 2.3.2 Severity of depression.

- Beck Depression Inventory (BDI)
- Patient Health Questionnaire-9 (PHQ-9)

### 2.3.3 Severity of anxiety.

- State-Trait Anxiety Inventory (STAI)
- Generalized Anxiety Disorder 7 (GAD-7)

### 2.3.4 Quality of life.

- Short Form-36 (SF-36)
- World Health Organization Quality of Life (WHOQOL)

### 2.3.5 Well-being and life satisfaction.

- Satisfaction with Life Scale (SWLS)
- Oxford Happiness Questionnaire (OHQ)

### 2.3.6 Stress levels.

- Perceived Stress Scale (PSS)
- Child-PSS (CPSS)

### 2.3.7 Suicidal ideation and behavior.

- Scale for Suicidal Ideation (SSI)
- Columbia-Suicide Severity Rating Scale (C-SSRS)

### 2.3.8 Stereotyped behaviors, communications, and social interaction.

- Gilliam Autism Rating Scale-Second Edition (GARS-2)

Other types of outcomes will be considered based on the research designs of the studies retrieved and included in the review. The effect measures for our primary outcomes will be categorized using Cohen's d classification, and we will also consider the significance level ($p < 0.05$).

## 2.4 Comparators

In this systematic review, we investigate the landscape of physical activity, exercise, and sports involvement within diverse age groups, encompassing children, adolescents, and individuals of varying ages. Three primary comparison groups including age groups of children and/or adolescents with or without disabilities will be considered:

1. not participating in any form of sport or organized physical activity/exercise, provide a baseline for understanding the consequences of inactivity.

2. engaged in sports or organized physical activities other than OCS, allowing for comparisons across different modes of physical participation.

3. exhibiting a level of expertise in OCS, or actively participating in OCS programs different from those of the experimental group. Thus, this cohort should exhibit differences in athletic proficiency, characteristics (e.g., competitive level, training volume), and practice settings, providing insights into the multifaceted aspects of OCS involvement.

As specifically regards intervention studies, we will include any type of comparison group, categorizing them as (i) passive, (ii) non-physically active and (iii) physically active and considering the eventual differences that may arise from using different comparison groups. Indeed, an overall issue is the Hawthorne effect, which has been shown to lead to larger effects when the comparator is a passive group, compared to active controls involved in any specific activity alternative to the intervention. Beyond this overall issue, specifically as concerns physical activity/sport interventions, various non-physically active (e.g., video lecture) and physically active (e.g., walking/stretching) control activities may generate a different expectancy for improvement than other activities (e.g., yoga/meditation), potentially leading to differential effects [33].

Collectively, these comparison groups enable a comprehensive examination of the effects of physical activity, sports, and exercise on mental health on or association with related outcomes. They offer valuable insights into the advantages, potential risks, and contextual factors associated with various levels and types of physical engagement.

## 2.5 Search strategy

A comprehensive search will be conducted across seven electronic databases, including Scopus, ERIC, PsycINFO, PubMed, Cochrane Library, Web of Science, and SPORTDiscus. The search strategy, which has been pilot-tested, is provided in Table 1. This strategy is devised based on the primary outcome variables of (1) OCS, (2) mental health, (3) the target population specifically children and adolescents with disabilities, and (4) the relevant study designs and terminologies. In cases where studies are not in the English language, inclusion will be considered if an English translation exists or can be obtained with the assistance of members of our review team. Solely peer-reviewed studies will be incorporated into the review, while grey literature, such as research reports, working papers, conference proceedings, and theses/dissertations, will be excluded during the initial search and screening phases. To provide a comprehensive search of the relevant articles, the snowball technique will be also applied.

## 2.6 Study records

During the initial screening phase, we will exclude duplicates and records from grey literature across various databases. This initial screening will be conducted prior to the commencement of the blinded review process and will be carried out by a single team member (referred to as YL). To facilitate this, we will utilize EndNote x9 [34], a reference management software.

Table 1. The search terms and strategies.

| Domain | Search terms |
|---|---|
| Olympic Combat Sports | (Olympic combat sports" OR "Olympic martial arts" OR boxing OR boxer* OR wrestling OR wrestler* OR taekwondo OR taekwondoist* OR judo OR judoka* OR judoist* OR fencing OR fencer* OR karate OR karateka* OR karateist* OR combat AND sports OR "martial arts" OR "martial combat" OR "combat competitions" OR "combat techniques" OR "combative sports" OR "combat training" OR "striking sports" OR "grappling sports" OR "submission sports" OR "self-defen* sports") |
| | AND |
| Mental Health | ("mental health" OR "mental illness" OR "psychological health" OR "psychiatric disorder" OR "behavioral health" OR "emotional well-being" OR "psychological well-being" OR "psychosocial well-being" OR "mental well-being" OR "mental disorder" OR "mental distress" OR psychopathology OR "mental wellness" OR "mental state" OR "mental resilience" OR "mental disability" OR "cognitive impairment" OR "neurological disorder" OR "psychological distress" OR "mental condition" OR "mental state" OR "mental dysfunction" OR emotion* OR depression* OR anger* OR anxiet* OR stress* OR happiness OR self-esteem* OR self-concept* OR "body image" OR motivation* OR "motivational factors" OR "motivational processes" OR "intrinsic motivation" OR "extrinsic motivation" OR "self-determination" OR "goal orientation") |
| | AND |
| Disability | (disabilit* OR disabled OR impairment* OR handicap* OR "chronic illness" OR "functional limitation*" OR "mobility limitation*" OR "activity limitation*" OR "participation restriction*" OR adapt* OR inclusive* OR "Special Education*" OR inclusion* OR "special need*" OR adaptive* OR Wheelchair* OR accessible* OR special* OR rehabilitation* OR supportive* OR autism* OR "autism spectrum disorder" OR "cerebral palsy" OR depression* OR "bipolar disorder*" OR schizophrenia* OR trauma* OR sclerosis* OR fibromyalgia* OR epileps* OR fatigue* OR "down syndrome" OR Parkinson OR Alzheimer OR "borderline personality disorder*") |
| | AND |
| Study design | (RCT) OR ("control* trial*") OR (quasi) OR (intervention*) or (longitudinal) or (cohort) or (prospective) or (cross-sectional) |
| | AND |
| Target Population | (child* OR school* OR youth* OR student* OR adolescent* OR adolescence* OR juvenile*) |

Subsequently, the same team member (YL) will upload the resultant list to Covidence [35], an online tool specifically designed for systematic literature reviews (SLRs). Within Covidence, the blinded review process will encompass title and abstract screening, full-text screening, study selection, data extraction, and risk of bias assessment. Covidence follows the PRISMA flow diagram for SLRs [36] and enables the distribution of studies among multiple reviewers. To ensure that all reviewers are well-versed in the procedures and to promote consensus among team members, several meetings will be convened before each stage of the review, including study screening, risk of bias assessments, and data extraction. A core group within the review team will serve as guides and provide support to fellow team members as they navigate the various stages of the review process.

## 2.7 Screening process

During the title and abstract screening, as well as the full-text screening phases, each study will undergo evaluation by two impartial and independent reviewers who are part of the review team. In instances where discrepancies arise between these independent reviewers, a third reviewer from the core group will step in to resolve any inconsistencies. Covidence will be utilized to distribute an equivalent number of studies among reviewers, with the distribution being randomized. In the first phase, titles and abstracts will be assessed for eligibility using a predefined decision tree, which is based on the inclusion and exclusion criteria anticipated to

be evident in either the title or the abstract. During the second phase, full texts will be scrutinized for eligibility based on the complete set of inclusion and exclusion criteria. Any reasons for the exclusion of studies at this stage will be meticulously documented according to categories of reasons for exclusion progressively refined in Covidence to ensure the best possible transparency. Following the full-text screening, one reviewer will undertake a thorough examination of the included studies to eliminate instances of duplicate reporting. Duplicate reporting refers to the situation where the results from the same sample are reported in multiple studies or where studies have been published more than once. To achieve this, various aspects of the studies will be compared, including authorship, study locations and settings, intervention details, study design, sample size, and demographic information [37]. If duplicate reporting is identified among the included studies, the reviewers will make efforts to pinpoint the primary study that was duplicated. In cases where the primary study cannot be discerned, preference will be given to the study with the longest follow-up period or the highest number of measurement time points [37, 38].

## 2.8 Data extraction

A data extraction template will be generated within Covidence and subjected to a preliminary trial before the actual data extraction phase. Two independent reviewers will be responsible for extracting data from each study. In cases where information or data are found to be missing or require clarification, the corresponding author of the respective studies will be contacted. To handle cases where data from figures in the included studies are not directly available, we will contact the corresponding authors of these studies and kindly request the raw data for review purposes. If the authors do not respond or the data remains unavailable, we will document these instances and consider them during the data synthesis process. If a response is not received before the completion of the data extraction process or if the reporting remains incomplete, the study will be excluded. Following the independent data extraction, the two reviewers will engage in a consensus procedure to resolve any disagreements and validate the accuracy of the extracted data. The data to be extracted will encompass the following components:

- Study and Intervention Details: This includes study design, a concise description of the study's intervention, details regarding the design and content of the intervention, a description of the control group's physical or non-physical activities, and information about the study setting.

- Participant Information: This section will capture sample size, the age of the participants (including age distribution by gender), gender distribution (with a breakdown by gender percentage), and disability type.

- Outcome Measures and Modifiable Factors: This component will encompass various outcome measures and modifiable factors.

- Timeframes: Information regarding the duration of the intervention (in weeks), the location of the intervention (country), the number of measurement time points, and the duration of follow-up (in weeks).

- Results Data: This section will encompass outcome data (mean values and measures of variance) as well as data related to modifiable determinants (mean values and measures of variance).

## 2.9 Data aggregation

To ensure the validity of our review's conclusions, our approach to aggregating data will be meticulously defined, considering the homogeneity, similarity, and consistency of the included studies. Before conducting meta-analyses, we will perform an exploratory analysis to assess the degree of statistical heterogeneity among studies. This will involve:

- Assessing Homogeneity: Using statistical tests and visual inspection of forest plots to identify potential outliers and the overall distribution of effect sizes. This step is critical for deciding whether a meta-analysis is appropriate and determining the most suitable model (random-effects or fixed-effects) for data aggregation.

- Similarity and Consistency Check: Evaluating the methodological and clinical similarities among studies to ensure comparability. This will include an examination of study designs, populations, interventions, outcomes, and measurement tools. Consistency will be assessed through sensitivity analyses, examining the impact of excluding studies with a high risk of bias or those that are significantly different in terms of design or population.

## 2.10 Risk of bias

To evaluate the methodological quality and internal validity of the included studies, a risk of bias assessment will be conducted across various study designs. Different assessment tools will be applied based on the specific study type. For RCTs, we will employ a modified version of the Rob 2.0 tool [39], as recommended by the Cochrane Collaboration. This tool encompasses the following domains: bias arising from the randomization process, bias due to deviations from the intended interventions, bias arising from missing outcome data, bias in the measurement of the outcome, and bias in the selection of the reported result. Non-RCTs will undergo a risk of bias assessment using a modified version of the ROBINS-I (Risk Of Bias In Non-randomized Studies–of Interventions) tool [40]. This tool considers the following domains: bias due to confounding, bias in the selection of participants, bias in the classification of interventions, bias due to deviations from intended exposures or interventions, bias due to missing data, bias in the measurement of outcomes, and bias in the selection of the reported result. Longitudinal and cross-sectional studies will be assessed using the National Institute of Health's Study Quality Assessment Tool for Observation Cohort and Cross-sectional Studies. The two independent reviewers responsible for data extraction from the respective studies will also perform the risk of bias assessment to ensure familiarity with the intricacies of each study. The risk of bias assessment will be carried out using forms created in Covidence, with the respective risk of bias tools serving as templates. After independent data extraction, the two reviewers will engage in a consensus procedure to resolve any disagreements and validate the accuracy of the assessment.

## 2.11 Data synthesis

Data extraction will result in a dataset comprising information from RCTs, CTs, and observational studies with control groups and interventional studies (RCTs and non-RCTs), encompassing populations both with and without disabilities. An overview table will be generated, presenting the general characteristics of the included studies, including details on methods, settings, modifiable determinants, sample attributes (such as size and age), and outcomes (comprising outcome measures, types of measures, number of measures, and measurement time points) [41]. Findings will be synthesized narratively to identify and compile a list of modifiable determinants and the settings in which they were investigated. Most of the data

extracted from the included studies are expected to be in continuous format. Whenever possible, meta-analytic techniques will be applied. Meta-analyses (MAs) will be carried out using both frequentist and Bayesian approaches, utilizing the JASP statistics software [42]. In the frequentist pairwise comparisons, the direct effect will be examined. This involves calculating the standardized mean difference (SMD) and its 95% confidence intervals (CIs) using post-intervention data while controlling for baseline differences. Heterogeneity will be assessed using Cochrane's Q, based on a $\chi^2$ test with consideration of the CI size relative to the degrees of freedom (df). Additionally, $I^2$ will be employed to quantify the extent of consistency across studies. Interpretation of heterogeneity will follow established benchmarks: $I^2 < 25\%$ signifies low heterogeneity, $25\% < I2 < 50\%$ indicates moderate heterogeneity, and $I^2 > 75\%$ suggests high heterogeneity [43, 44]. Bayesian methods will be used for the MAs with random effects models. Bayesian meta-analysis offers advantages, including the incorporation of prior knowledge, nuanced conclusions beyond p-values, and assessment of result plausibility within a probability range [45, 46]. Gibbs sampling of the Markov Chain Monte Carlo algorithm will be applied using JASP [42]. To gauge the probability of publication bias, the RoBMA (Robust Bayesian Meta-analysis) extension in JASP will be employed. This approach facilitates state-of-the-art publication bias-adjusted meta-analysis [47, 48]. Bayesian model averaging will be utilized, considering multiple plausible models, and addressing concerns about model selection [49]. Additionally, RoBMA offers benefits such as continuous scale evidence quantification, avoidance of accumulation bias, and mitigation of estimation issues through prior distributions. Prior specifications and models will be used, with the modification of excluding fixed-effects models [47].

## 2.12 Quality assessment

Following the completion of meta-analyses, we will assess the quality of the evidence for each key outcome using the GRADE methodology [50]. This systematic approach will evaluate the evidence based on the risk of bias, consistency, directness, precision, and publication bias. The quality of evidence will be categorized as high, moderate, low, or very low. These categorizations will directly inform our confidence in the effect estimates provided for each outcome. Should any discrepancies arise in our GRADE assessments, they will be resolved through discussion among the research team or by consulting a third reviewer. The outcomes of these assessments will be succinctly summarized in a 'Summary of Findings' table. This table will provide clear, evidence-based guidance regarding the strength of evidence supporting the impact of Olympic combat sports on mental health outcomes for children and adolescents with disabilities.

## 2.13 Moderation analysis

Data will be discussed also in relation to the context of various settings, considered not merely as the background of the intervention but as a relevant potential moderator of effects/associations, and the quality of evidence incorporated into the review [51, 52]. This analysis will specifically consider the following types of moderators, contingent on the availability of a sufficient number of studies to facilitate such analysis:

- Individual-level moderators: These include factors such as gender, age, type or level of disability, and level of OCS expertise. In interventional studies, the level of expertise is assessed to determine if it acts as a potential moderator of intervention effects rather than being the focus, which contrasts with its role in cross-sectional studies.

- Task-level moderators: Applicable solely to intervention studies, this category encompasses variations in dosages or types of OCS practice. It aims to analyze how different approaches to OCS practice may influence the outcomes of interventions.

- Context-level moderators: This pertains to the physical and social environment, the setting of the intervention, and the mode of its delivery. The context in which an intervention occurs can significantly affect its effectiveness and the nature of its impact on participants.

- Outcome-level moderators: Different domains of mental health outcomes will be examined focusing on healthy participants across the lifespan. This will allow for a nuanced understanding of how various mental health outcomes may be differently affected by interventions.

- Study-level moderators: Factors such as the type of comparator used in studies (e.g., passive; physically active, involving another type of physical activity or sport; non-physically active, involving a cognitively or socially challenging activity without a physical activity component) and the quality of the studies (lower vs. higher, or at least moderate quality) will be considered. These moderators are critical for assessing the relative effectiveness of OCS interventions and the reliability of the evidence.

By incorporating these types of moderators into the analysis, the review aims to provide a comprehensive understanding of the factors that may moderate the effects of OCS interventions on various outcomes. This approach recognizes the complexity of intervention effects and the importance of a nuanced analysis that considers individual, task, context, outcome, and study-level variables.

## 2.14 Ethics and dissemination

This protocol outlines the systematic approach to conducting a series of SLRs and MAs to comprehensively investigate the association between OCS and mental health in children and adolescents. The results obtained from these studies will be shared through peer-reviewed publications and, when feasible, academic conferences. Since no primary data collection is involved, ethical approval is not necessary for this research endeavor.

## 3 Discussion

This systematic review will endeavor to unravel the relationship between OCS and various dimensions of mental health, building upon the piecemeal evidence from primary studies [10]. By meticulously categorizing and distinguishing between interventional and non-interventional studies in the unique context of children with disabilities, we will aim to illuminate novel avenues of research that will significantly impact the sports sector's policy-making landscape. In doing so, we acknowledge that while existing studies, such as those highlighted by Te Velde et al. [27], have hinted at the positive effects of sports on the mental health of individuals with disabilities, a comprehensive systematic review is imperative. Such a review is essential not only to synthesize the available evidence and draw initial conclusions that can inform the development of future research but also to fill evidence gaps and further our understanding of the OCS-mental health relation in this special population. This review will represent a distinctive contribution to the existing body of literature in several ways. First and foremost, it will offer a comprehensive examination of OCS in the context of children and adolescents with disabilities, an area that has received limited attention in previous reviews. Our focus on diverse disability types, sports, and sports contexts, combined with a meticulous analysis of various facets of the mental health construct, will distinguish this review from earlier, more generalized

studies. The rigorous inclusion and exclusion criteria, robust search strategy, and careful assessment of study quality will enhance the reliability and validity of the findings.

However, the present review may also highlight differences among studies that can render the evidence base useful and generalizable for policymakers. Such differences may especially regard the individual (disability type), sport task-related (intervention type and length), and context-related moderators (competition level, delivery, physical and social context) that may amplify or buffer the effects of OCS practice on mental health outcomes or their non-interventional association. The attention to individual, task-related and context-related moderators of the OCS-mental health relation will provide practical implications and pave the way for research on what works, for whom and under which circumstances applied to OCS interventions with complex implementation chains for youths with disability [53]. The analyses of our systematic review can provide evidence that furthers our understanding of how OCS interventions are intended to work and how complexity in both the interventions and their environmental context may explain differences in their impact on mental health [54].

This trade-off will aim to prioritize the depth of analysis within our chosen scope. While the introduction to this review will emphasize the potential benefits of OCS for children with disabilities, it will be essential to acknowledge the presence of potential negative aspects. Indeed, the pursuit of sports, even in an inclusive and supportive environment, can pose challenges and risks for some individuals, including physical demands, competitiveness, and the potential for injury. Acknowledging these nuances will be important, and the review will consider these aspects with a balanced perspective. In conclusion, this systematic review will aim to provide a holistic and nuanced understanding of the relationship between OCS and mental health in children with disabilities. It will recognize the potential benefits while also addressing the inherent complexities and potential drawbacks. Through careful analysis, it will aim to equip policymakers and stakeholders in the sports sector with valuable insights to better support this special population.

## 4 Conclusion

We have intensified our efforts to delineate the novelty and methodological comprehensiveness of our review on OCS and mental health in children and adolescents with disabilities. This protocol pioneers an investigation into a significantly underexplored area, merging the dynamic fields of sports science and mental health with a focus on a vulnerable population. Our methodological approach is targeted to set a benchmark for rigor and comprehensiveness in systematic reviews. By employing both traditional and innovative analytical techniques, including Bayesian meta-analysis and sensitivity analyses, we not only adhere to but also advance the methodological standards for systematic reviews. This dual emphasis on novelty and methodological richness is pivotal, as it not only underscores the unique contribution of our study to the literature but also ensures the reliability and validity of our findings. Furthermore, the rationale for publishing this protocol paper is solidified by its potential to guide future research, policy formulation, and practice, filling a critical gap in the existing literature.

## Supporting information

**S1 Checklist. Reporting checklist for protocol of a systematic review and meta analysis.**
(DOCX)

**S1 File. Reporting checklist for protocol of a systematic review and meta analysis.** This is the PRISMA-P checklist detailing the reporting items and page references for the systematic review protocol.
(DOCX)

## Author Contributions

**Conceptualization:** Youngjun Lee, Janet Hauck, Simone Ciaccioni.

**Investigation:** Youngjun Lee, Laura Capranica, Caterina Pesce, Flavia Guidotti, Valentin Benzing, Janet Hauck, Simone Ciaccioni.

**Methodology:** Youngjun Lee, Caterina Pesce, Janet Hauck, Simone Ciaccioni.

**Supervision:** Janet Hauck, Simone Ciaccioni.

**Validation:** Laura Capranica, Caterina Pesce.

**Writing – original draft:** Youngjun Lee.

**Writing – review & editing:** Youngjun Lee, Laura Capranica, Caterina Pesce, Flavia Guidotti, Valentin Benzing, Janet Hauck, Simone Ciaccioni.

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
