## [Decision Letter · Decision Letter 0]

2 Aug 2024

PONE-D-24-12098Olympic combat sports and mental health in children and adolescents with disability: A protocol paperPLOS ONE

Dear Dr. Hauck,

Thank you for submitting your manuscript to PLOS ONE. After careful consideration, we feel that it has merit but does not fully meet PLOS ONE’s publication criteria as it currently stands. Therefore, we invite you to submit a revised version of the manuscript that addresses the points raised during the review process.

**Dear author, as you can see, although the reviewers recognized the merit of the proposal, they had doubts regarding the submitted protocol. To submit a new version of this protocol, all questions must be answered in a response letter and changes must be highlighted in the manuscript.**

We look forward to receiving your revised manuscript.

Kind regards,

Leonardo Vidal Andreato, PhD

Academic Editor

PLOS ONE

Journal Requirements:

Reviewers' comments:

Reviewer's Responses to Questions

**Comments to the Author**

1. Does the manuscript provide a valid rationale for the proposed study, with clearly identified and justified research questions?

Reviewer #1: Yes

Reviewer #2: Yes

Reviewer #3: Yes

2. Is the protocol technically sound and planned in a manner that will lead to a meaningful outcome and allow testing the stated hypotheses?

Reviewer #1: Yes

Reviewer #2: Yes

Reviewer #3: Yes

3. Is the methodology feasible and described in sufficient detail to allow the work to be replicable?

Reviewer #1: Yes

Reviewer #2: Yes

Reviewer #3: Yes

4. Have the authors described where all data underlying the findings will be made available when the study is complete?

Reviewer #1: No

Reviewer #2: Yes

Reviewer #3: Yes

5. Is the manuscript presented in an intelligible fashion and written in standard English?

Reviewer #1: Yes

Reviewer #2: Yes

Reviewer #3: Yes

6. Review Comments to the Author

You may also provide optional suggestions and comments to authors that they might find helpful in planning their study.

**Reviewer #1:** The objective of the study is, through a systematic review protocol, to outline the rationale and methodology for investigating how participation in Olympic combat sports influences the mental health outcomes of this population. The study is characterized as Reviewing Study Protocols.

Although I recognize that the study plays an excellent role considering methodological rigor and quality from the search strategy, screening process, data extraction, data aggregation, analysis of risk of bias and methodologic quality, and data analysis (including possible meta-analysis), it was not applied in a real situation. This aspect is important, especially because the chosen topic is very specific - Olympic combat sports and mental health in children and adolescents with disability. There is a range of studies on combat sports, but when we stratify into children and adolescents this number decreases a lot and then when we stratify into disabilities and then mental health, even more so. The fact is that I'm not sure if the excellent proposed protocol is useful and can be used in practice due to the very low number of studies on this specific subject. Therefore, I strongly recommend that authors use the protocol and test its usability. Only then will we be able to see the importance of study for science.

**Reviewer #2:** The manuscript is not original study and not exactly review, it is proposition for kind of procedure. The topic is important (mental health). The weakness is the selection of literature. There are a lot of self-citations.

**Reviewer #3:** The authors demonstrated an interesting protocol for the systematic literature review regarding Olympic combat sports and mental health parameters among children and adolescents with disabilities. In addition, considering the robust, thorough, and high-quality methodological aspects, the future study will have the potential to be published in some of the most prestigious scientific journals in the area of sports sciences and applied exercise psychology. For these reasons, a minor revision of this study protocol is recommended! However, the stated comments and objections should improve the current version of the manuscript.

The suggested study title: Olympic Combat Sports and Mental Health in Children and Adolescents with Disabilities: A Protocol for Systematic Review.

Regarding the rationale of this investigation, it is recommended to emphasize potential innovative aspects of the future systematic review referring to Olympic combat sports and mental health. Expressly, will the literature synthesis approach only confirm the obtained results of the existing studies, or several new findings will be provided? Please report it, if possible.

The name of the subheading "types of studies" is entirely inappropriate. The authors should highlight that this subheading is related to the "study design"! In addition, do controlled trials (CTs) pertain to non-randomized controlled investigations? It is highly desirable to employ suggested terminology across the whole manuscript. Lastly, the text concerning the last claim in this subheading is redundant and should be presented within the first statement.

Please indicate whether the comparison groups will include children and adolescents with disabilities or their typically developing counterparts. Please highlight the missing facts!

How do you intend to extract data from the figures across the included studies? Please specify.

The text concerning certainty of evidence (GRADE assessment) must be presented as a separate subheading. Please provide these corrections!

Based on the relevant findings, will the future systematic review have significant practical implications? Please emphasize and discuss!

The manuscript contained several confusing statements and unsuitable expressions (lines 80-82; line 221). Please make the text fluent and improve your terminology!

7. PLOS authors have the option to publish the peer review history of their article (what does this mean?). If published, this will include your full peer review and any attached files.

Reviewer #1: No

Reviewer #2: **Yes: **Dariusz Boguszewski

Reviewer #3: **Yes: **Patrik Drid

---

## [Author Response · Author response to Decision Letter 0]

11 Sep 2024

Dear Editor and Reviewers,

Thank you sincerely for your valuable time and constructive comments and suggestions. We have considered all the comments carefully and revised our manuscript accordingly. Responses to the specific comments from reviewers and the editor can be found in the attached response to reviewers' document. We have addressed each point thoroughly to ensure clarity and completeness. Thank you.

---

## [Decision Letter · Decision Letter 1]

28 Oct 2024

Olympic combat sports and mental health in children and adolescents with disability: A protocol paper

PONE-D-24-12098R1

Dear Dr. Hauck,

We’re pleased to inform you that your manuscript has been judged scientifically suitable for publication and will be formally accepted for publication once it meets all outstanding technical requirements.

Kind regards,

Leonardo Vidal Andreato, PhD

Academic Editor

PLOS ONE

Additional Editor Comments (optional):

Reviewers' comments:

Reviewer's Responses to Questions

**Comments to the Author**

1. Does the manuscript provide a valid rationale for the proposed study, with clearly identified and justified research questions?

Reviewer #3: Yes

2. Is the protocol technically sound and planned in a manner that will lead to a meaningful outcome and allow testing the stated hypotheses?

Reviewer #3: Yes

3. Is the methodology feasible and described in sufficient detail to allow the work to be replicable?

Reviewer #3: Yes

4. Have the authors described where all data underlying the findings will be made available when the study is complete?

Reviewer #3: Yes

5. Is the manuscript presented in an intelligible fashion and written in standard English?

Reviewer #3: Yes

6. Review Comments to the Author

You may also provide optional suggestions and comments to authors that they might find helpful in planning their study.

Reviewer #3: The authors have presented an intriguing protocol for a systematic literature review focusing on mental health parameters among children and adolescents with disabilities in Olympic combat sports. Considering the robust and meticulous methodological approach, the study holds promise for publication in leading scientific journals specializing in sports sciences and applied exercise psychology. Therefore, I recommend acceptance.

7. PLOS authors have the option to publish the peer review history of their article (what does this mean?). If published, this will include your full peer review and any attached files.

Reviewer #3: **Yes: **Patrik Drid

---

## [Editor Report · Acceptance letter]

15 Nov 2024

PONE-D-24-12098R1 

PLOS ONE

Dear Dr. Hauck, 

I'm pleased to inform you that your manuscript has been deemed suitable for publication in PLOS ONE. Congratulations! Your manuscript is now being handed over to our production team.

Kind regards, 

on behalf of

Dr. Leonardo Vidal Andreato 

Academic Editor

PLOS ONE